# Cyclic Fatigue of Different Reciprocating Endodontic Instruments Using Matching Artificial Root Canals at Body Temperature In Vitro

**DOI:** 10.3390/ma17040827

**Published:** 2024-02-08

**Authors:** Sebastian Bürklein, Paul Maßmann, Edgar Schäfer, David Donnermeyer

**Affiliations:** 1Central Interdisciplinary Ambulance in the School of Dentistry, University of Münster, 48149 Münster, Germany; sebastian.buerklein@ukmuenster.de (S.B.); p_mass02@uni-muenster.de (P.M.); eschaef@uni-muenster.de (E.S.); 2Department of Periodontology and Operative Dentistry, University of Münster, 48149 Münster, Germany

**Keywords:** amplitude, cycles to failure, dynamic testing, lifetime, NiTi, reciprocating motion

## Abstract

Reciprocating motion expands the lifetime of endodontic instruments during the preparation of severely curved root canals. This study aimed to investigate the time to fracture (TTF) and number of cycles to failure (NCF) of different reciprocating instruments (*n* = 20 in each group) at body temperature using a dynamic testing model (amplitude = 3 mm). Reciproc Blue (RPB), size 25/.08, WaveOne Gold (WOG) 25/.07, Procodile (Proc) 25/.06, R-Motion (RM_06) 25/.06 and R-Motion (RM_04) 30/.04 instruments were tested in their specific reciprocating motion in artificial matching root canals (size of the instrument ± 0.02 mm; angle of curvature 60°, radius 5.0 mm, and centre of curvature 5.0 mm from apical endpoint). The number of fractured instruments, TTF, NCF, the and lengths of the fractured instruments were recorded and statistically analysed using the Chi-Square or Kruskal–Wallis test. Both TTF (median 720, 643, 562, 406, 254 s) and the NCF (3600, 3215, 2810, 2032, 1482 cycles) decreased in the following order RM_06 > RPB > RM_04 > Proc > WOG with partially significant differences. During testing, only six RM_06 instruments fractured, whereas 16/20 (RPB), 18/20 (Proc), and 20/20 (RM_04, WOG) fractures were recorded (*p* < 0.05). Within the limitations of the present study, blue-coloured RPB and RM instruments exhibited a significantly superior cyclic fatigue resistance compared to SE-NiTi and Gold-wire instruments. Heat treatment, cross-sectional design and core mass significantly influenced the longevity of reciprocating instruments in cyclic dynamic testing.

## 1. Introduction

Root canal treatment involves the removal of (infected) tissues from the root canal system of a tooth, which is usually accomplished using specialised stainless steel or nickel–titanium (NiTi) instruments [1,2]. NiTi instruments are widely used due to their unique mechanical properties, such as high flexibility and pseudoelasticity [3]. However, these instruments can fail due to cyclic fatigue, which occurs when they are subjected to repetitive cycles of loading and unloading during use. Especially in severely curved root canals, the instruments are heavily stressed depending on the degree and radius of the curvature.

The cyclic fatigue of NiTi instruments is a complex phenomenon influenced by various factors, such as instrument design, taper, cross-sectional geometry, core mass, surface and heat treatment [4,5,6,7,8].

Understanding the mechanisms of cyclic fatigue and identifying ways to improve the durability of NiTi instruments seem essential for achieving predictable and successful root canal treatment outcomes. Cyclic fatigue testing is a common method used to evaluate the mechanical properties of endodontic instruments that is typically standardised using specific parameters to ensure consistency and accuracy across different studies. Nonetheless, a recent publication highlighted the great variability in the available studies and questioned their comparability [9]. Authors recommended that the cyclic fatigue resistance should be given by manufacturers who must follow ADA Specification No. 28 (Council on Dental Materials & Devices 2013) [10] and the ISO Specification 3630-1:2008 (International Organization for Standardization 2012) [11] which per se have limited clinical relevance.

In a similar sense, better standardisation was recommended recently, as the instruments were subjected to completely different trajectories during testing due to their different sizes and tapers when using ISO and/or ADA standards [12]. The tests should at least be performed at body temperature using a dynamic approach [9].

Aiming at the comparability of the instruments either set into a pure unidirectional rotary motion or into a specific reciprocating motion, the parameter number of cycles to failure (NCF) was introduced instead of time to failure (TTF) to address different rotational speed and kinematics. It is important to understand that the assessment of the two parameters represents different aspects of cyclic fatigue performance, and NCF represents a purely mathematical parameter that is calculated by multiplying the rotational speed with the time elapsed until fatigue fracture occurs. The NCF provides information about the overall durability of the instrument, while the time to fracture provides information about how long the instrument can be used before it fails.

Reciprocation is a movement strategy used in endodontic instrumentation that involves alternating clockwise and counterclockwise rotation of the instrument. Compared to continuous rotation, reciprocation has been shown to reduce the occurrence of cyclic fatigue in endodontic NiTi instruments [13]. This is because reciprocation creates less stress on the instrument than continuous rotation, reducing the likelihood of fatigue failure. Additionally, the alternating motion of reciprocation avoids extensive stress on the instruments when a sectional apical binding occurs because the angles always stay below critical values for torsional fractures due to the defined unwinding action [14].

Overall, reciprocation movement is a promising strategy for reducing the incidence of cyclic fatigue in endodontic NiTi instruments and may offer a potential solution for improving their durability and performance. Even in clinical studies, the fracture incidence of reciprocating instruments is very low and must be declared as rare [15,16,17]. Other crucial parameters were the core diameter [18], the cross-sectional design [19] and the surface treatment of endodontic instruments [20].

The aim of this study was to compare the cyclic fatigue resistance of different reciprocation instruments with different designs, tapers, sizes and heat treatments in matching artificial severely curved root canals concerning NCF and time to fracture in a newly developed testing device guaranteeing high standardisation [12].

The null hypothesis was that cyclic fatigue resistance, i.e., TTF and NCF, is equal for all the instruments tested.

## 2. Materials and Methods

The experimental setup presented in the following aimed to evaluate cyclic fatigue of different engine-driven reciprocating NiTi instruments offering different designs, tapers and sizes using standardised matching artificial root canals guaranteeing the same trajectories.

### 2.1. Sample Size Calculation

Prior to the evaluation of the main study, a preliminary investigation under identical conditions was performed using five instruments in each group. Mean values and standard deviations revealed an effect size >1.0 (Cohen’s d). A markedly lower effect size of 0.4 served for calculation using G*Power 3.1 (Heinrich Heine University, Düsseldorf, Germany). Using the parameters of an alpha (α) level of 0.05 (5%), a beta (β) level of 0.20 (20%) (i.e., power = 80% at a 5% significance level) and an effect size of 0.4, the sample size was 16 in each group. In total, 20 instruments in each group were used (total = 100).

### 2.2. Artificial Canals

Based on the findings of a previous study, artificial root canals matching the dimensions of the instruments investigated were designed using Geomagic Freeform software (v2022.0.34, 3D Systems, McLean, Fairfax, VA, USA) and transferred into a CoCr model using a digital metal laser sintering (DMLS) method (Infinident Solutions, Darmstadt, Germany) [12]. In total, four different artificial canals were used for the cyclic fatigue tests, corresponding to the instruments used in this study: (i) 25/.06, constant taper for Procodile (=Proc) and R-Motion 25/.06 (=RM_06), (ii) 25/.07, variable taper for WaveOne Gold Primary (=WOG), (iii) 25/.08, variable taper for Reciproc Blute (=RPB), and (iv) 30/.04, constant taper for R-Motion 30/.04 (=RM_04). All diameters of artificial canals were increased by 0.02 mm, representing the tolerances permitted in the manufacturing process of endodontic instruments [11].

### 2.3. Testing Device

A dynamic testing approach in an incubator set at body temperature (37 °C) was performed. Glycerine oil warmed to body temperature served as a lubricant. Continuous refreshment during testing guaranteed a friction-free reciprocation motion. The artificial canals were covered with tempered glass to prevent the instruments from slipping out of the matching tubes [12].

An eccentric mount 1.5 mm from the central axis (total amplitude = 3 mm) was fixed on a continuously rotating drive disk to simulate dynamic up and down movements (picking movement, stroking) during root canal treatment with a speed set to 1 cycle/2 s (=0.5 Hz). To ensure a fixed and standardised positioning of the handpiece and the instruments, a 3D-printed and assembled handpiece holder, which was specially developed and manufactured for this study, was used. The test machine allowed the components to be freely adjusted in all directions, guaranteeing exact axial insertion without any preload of the instruments.

All instruments (Figure 1) were tested according to the manufacturer’s instructions—rotational speed and torque settings were set to the demanded values using a torque-controlled endodontic motor (VDW silver, VDW, Munich, Germany):

**Reciproc Blue 25/.08 (RPB)** (VDW, Munich, Germany)—variable taper, S-shaped cross-section, Blue heat treatment; motion kinematics: “reciprocation all” mode (approximately 150° CCW (= counter-clockwise)/30° CW (=clockwise); 10 cycles/second, resulting in about 300 rpm (= rotations per minute).

**Procodile 25/.06 (Proc)** (Komet, Lemgo Germany)—constant outer instrument taper, innovative regressive core taper, S-shaped cross-section, Super-Elastic (=SE)-NiTi; motion kinematics: “reciprocation all” mode.

**WaveOne Gold Primary 25/.07 (WOG)** (Dentsply Maillefer, Ballaigues, Switzerland)—variable taper, parallelogram cross-sectional design with either one or two active cutting edges, Gold heat treatment; motion kinematics: “Wave One” mode (approximately 170° CCW (=counter-clockwise)/50° CW (=clockwise); 10 cycles/second, resulting in about 300 rpm.

**R-Motion 25/.06 (RM_06)** (FKG Dentaire SA, La Chaux de Fonds, Switzerland)—constant tapered, triangular cross-sectional design, proprietary heat treatment, electro-polished surface; motion kinematics: “reciprocation all” mode.

**R-Motion 30/.04 (RM_04)** (FKG Dentaire SA, La Chaux de Fonds, Switzerland)—constant tapered, triangular cross-sectional design, proprietary heat treatment, electro-polished surface; motion kinematics: “reciprocation all” mode.

The tests were recorded with a microscopic camera using AMcap software (AMcap, v3.0.1.7; NCH Software, Greenwood, CO, USA). The automatic lifting device and the endodontic motor were started simultaneously. When an instrument fractured, the testing machine and then the video were stopped. The final temperature was noted in the experimental protocol and the video was saved.

The start and end time of the experiment were determined based on the video recording (Movies and TV App, Microsoft Corporation, Redmond, WA, USA, version 10.20112.10111.0). The start time was defined as the moment when the endodontic instrument first reaches the working length. The end time was defined as the time at which the instrument fractured. All the tests were limited to 720 s.

The cycles to fracture were determined based on the time of instrument failure:NCF = time to fracture [s]/60 × *rpm*
*NCF* = *number of cycles to fracture*; *rpm* = *rotations per minute*

### 2.4. Analysis of the Fractured Instruments

The lengths of the fractured instrument tips were measured with a digital calliper (Mitutoyo 500-196-30 Absolute AOS Digimatic; Mitutoyo Corporation, Kawasaki, Japan).

To assess the fracture mode, five randomly selected fragment surfaces in each group, if available, were examined using a laser scanning microscope (Keyence VK-X1000, Keyence Corporation, Osaka, Japan) at a 500-fold magnification to exclude a torsional fracture. For the examination, the fracture surfaces were cleaned with 70% alcohol and embedded in a silicone block.

Fractographic analysis of cross-sectional images of fractured instruments reveals the characteristic fatigue fracture pattern of crack initiation, propagation and catastrophic fracture with pitting. Pitting over the entire fracture surface is typical of cyclic fatigue fractures, while torsional fractures exhibit circular wear marks on the fracture surface and pitting only near the centre of rotation [21,22].

### 2.5. Statistical Analysis

Statistical analysis was performed using SPSS 20 software (v28.0.1.1, SPSS Inc., Chicago, IL, USA). The obtained data were checked for normal distribution using Kolmogorov–Smirnov and Shapiro–Wilk tests. Non-normally distributed values were evaluated using the Kruskal–Wallis test with Bonferroni correction. Chi-square test served for statistical analysis of the number of fractured instruments. The significance level was set at *p* = 0.05.

## 3. Results

The number of fractured instruments, the time and cycles to fracture and the fracture lengths of the instruments are listed in Table 1. Median and corresponding lower and upper 95% confidence intervals were given because not all groups had normally distributed data (Kolmogorov–Smirnov and Shapiro–Wilk tests).

### 3.1. Number of Fractured Instruments

Only six RM_06 instruments fractured, whereas 16 RPB and 18 Proc files and all WOG and RM_04 fractured. The differences were significantly different in the increasing number of failures: RM_06 < RPB, Proc < WOG, RM_04 (*p* < 0.05).

### 3.2. Time to Fracture and NCF 

Both TTF and NCF showed similar significant differences (Figure 2 and Figure 3). The highest fatigue resistance was obtained using RM_06 followed by PR, RM_04, Proc and WOG. The median from TTF even reached 720 s in the RM_06 (CI (=confidence interval): 655–724) group representing the end of the recording time. However, the TTF between RM_06 and RPB (643; CI: 590–662) did not differ significantly (*p* > 0.05). RM_04 (562; CI: 527–589) formed the next group with significant differences to RM_06 (*p* < 0.05) but not to RPB (*p* > 0.05). Procodile (406; CI: 369–488) and WOG (254; CI: 242–262) differed significantly from each other and all other groups (*p* < 0.05), except Proc and RM_04, where no significant difference was obtained (*p* > 0.05).

The significances between TTF and NCF varied slightly due to the higher speed in WOG reciprocation mode that led to a higher number of cycles during instrumentation/testing. Thus, WOG and Proc did not show significantly different values (*p* > 0.05), whereas all other significances did not change (Table 1).

### 3.3. Lengths of the Fractured Instruments

The fragment lengths of RPB (median = 3.55 mm; CI: 2.62–3.88) and RM_04 (3.03 mm; CI: 2.81–3.20) instruments were significantly longer than those of WOG (2.15 mm; CI: 2.17–2.48) (*p* < 0.05), whereas Proc (2.85 mm; CI: 2.63–2.99) and RM_06 (2.42; CI: 2.24–3.17 mm) did not differ significantly from the other groups (*p* > 0.05).

### 3.4. Fractographic Analysis

Five randomly selected fractured instruments from each group were analysed for typical signs of a cyclic fatigue fracture. These surfaces revealed typical areas of fatigue propagation, steady crack origin, and growth. Stress propagation is characterised by striations and catastrophic failure by rippled areas [12]. In all the tested instruments, cyclic fatigue fracture mode was verified.

## 4. Discussion

The study assessed the cyclic fatigue resistance of various reciprocating endodontic NiTi instruments. Thus, instruments representing different and specific designs, heat treatments and processing procedures were included, as already described. Reciproc Blue and WaveOne Gold represent the most widespread and market-leading reciprocating instruments with a Blue and Gold wire heat treatment. Procodile is a representative of conventional SE-NiTi instruments, and R-Motion (unique “Blue-like” heat treatment) is the most recent instrument to be launched on the market.

The present results showed superior cyclic fatigue resistance of RPB and RM instruments compared to conventional SE-NiTi and Gold-wire instruments. Thus, the null hypothesis was rejected. R-Motion instruments had the highest cyclic fatigue resistance followed by Reciproc Blue, Procodile, and WaveOne Gold (*p* < 0.05).

The present results corroborate current findings showing the superior performance of proprietary heat treatments resulting in blue-coloured instruments [23,24,25]. Heat treatments of NiTi alloys strongly influence the martensitic/austenitic transformation behaviour, favouring a different arrangement of the crystalline structure and a higher percentage of martensitic transformation [26]. Whereas R-Motion instruments (proprietary heat treatment) have an austenite finish temperature (A_f_) just below body temperature, between 32° and 35 °C [26], Reciproc Blue (Blue wire) and Wave One Gold (Gold wire) have an A_f_ temperature at body temperature (33–38 °C) and markedly above (about 47–51 °C), respectively [27]. SE-NiTi Procodile instruments show an A_f_ temperature of about 21–22 °C [28].

Surprisingly, the Gold-wire instruments (WOG) had a shorter lifetime compared to SE-NiTi instruments (Procodile), even though heat treatment usually increases cyclic fatigue resistance [29]. However, all instruments differed concerning many parameters, i.e., taper and cross-sectional design (Figure 4). WOG has a fixed taper from D1–D3, yet a progressively decreasing percentage tapered design from D4–D16, which serves to preserve middle and coronal root dentin. In the current setup, the centre of the curvature was 5 mm short of the apical endpoint. Thus, the outer diameter of the instruments differed: WOG represented 0.58 mm, whereas Procodile measured 0.55 mm at the same level. This may lead to an increased distance from the neutral fibre in the middle to the surface of the instrument leading to higher tension and crack propagation. In general, S-shaped cross-sections are claimed to have superior cyclic fatigue resistance [19]. Apart from that, the effect of core mass itself on dynamic cyclic fatigue is still controversial and is not the only factor influencing instrument breakage [8,30,31].

Obviously, all the instruments had a different cross-sectional design, core mass, outer diameter and taper, as seen in Figure 4. Each instrument was embedded in resin and cut at 6 mm from the tip (Figure 4). The inner core diameter (dotted line) and external dimension (full circle) were marked. When comparing both R-Motion instruments with an identical cross-sectional design, RM_06 (size # 25, taper 0.06) performed significantly better with regard to cyclic fatigue resistance compared to the instrument with a size of 30/.04 (*p* < 0.05).

At the centre of the curvature 5 mm from the apical endpoint, the instrument with a size of 30/.04 offered a diameter of 0.5 mm, whereas the RM_25/.06 had a 0.55 mm diameter. This finding contradicts a recent study that showed a superior performance of smaller NiTi wires when tested according to ISO [32]. In the present study, the instruments with larger core masses performed best.

Nonetheless, it has to be taken into consideration that the tests took place in matching artificial root canals that compensated the markedly lower curvatures of smaller instruments in parallel tubes [12,33]. With regard to the core diameter of all instruments tested in the present study, they had the following sizes in a decreasing sequence: RPB > WOG > RM_06 > RM_04 > Proc, which did not correspond to any of the parameters investigated. Consequently, the inner and outer core diameter did not represent predictive parameters for cyclic fatigue—other aspects outweighed.

The more decisive parameter when compared to WOG instruments may be related to the different motion kinematics. The reciprocation motion in general improves the cyclic fatigue resistance of endodontic instruments [34,35]. However, the Wave One mode generates larger angles in both directions (CCW/CW) within the same number of cycles per second, and thus has a higher rotational speed (350 vs. 300 rpm). Consequently, more cyclic stress may be applied to the instruments [25]. The assessment of NCF should overcome the different speed and kinematic settings, but the statistical evaluation of both TTF and NCF only slightly differed. Due to the different settings, only the significant difference obtained between WOG and Proc in TTF disappeared when NCF was analysed.

The superior dynamic cyclic fatigue resistance of R-Motion instruments may be attributed to both the heat treatment [26] and its specific electro-polished surface. This unique processing step consists of a final surface treatment that allows for controlled electrochemical removal of surface material, resulting in a smoother surface with fewer defects and residual surface stress [36,37].

The length of the fractured instruments was also assessed. The WOG instrument, which offered the shortest durability during cyclic fatigue testing, showed the significantly shortest fragment length. Despite the significant differences in the obtained data, the clinical relevance of this measurement remains unclear. To extract something meaningful from the fact that all instruments fractured between about 2.15 and 3.55 mm—meaning within 1.4 mm—seems somehow sophistically from the clinical point of view [33]. Avoidance of separating instruments seems more important than discussing the lengths of the fragments inside the root canal.

The relevance of laboratory studies for the evaluation of the properties and performance of endodontic instruments seems limited [9]. Martins et al. already questioned the necessity of the laboratory evaluation of different instruments’ properties by special testing devices [38]. The authors proposed to use a finite element analysis and/or a multimethod research approach instead that may lead to superior data collection, analysis, and interpretation of results, which when associated with a reliable confounding factor control and proper study designs, may be helpful tools and strategies to increase the reliability of the outcomes [38]. However, the limitations of the testing devices—even in a multimethod model—were based on the same or similar methods that were criticised in the mentioned publication. Limitations of a study do not become better when performed solitarily or in a combination with several methods without eliminating the inherent limitations. Generally speaking, conclusions and the transfer to clinical conditions always depend on the quality of the methods used and their specific clinical relevance. It can be stated that each mechanical test with its specific settings or parameters is somehow unique.

Finite element analysis may play an important role in the future, but it has some limitations, too. The analysis has to take numerous parameters into consideration, and validation of experimental data and fatigue data is indispensable. A recent study focused on the cyclic fatigue resistance of NiTi endodontic instruments, and the performed FE analysis elucidated that further research is needed to evaluate further effects such as specific surface and heat treatment [39]. In the aforementioned study based on the ISO standard, it was not possible to simulate the NCF using FEA to determine an estimated fatigue life [39]. Thus, the implementation of the dynamic approach with the cyclic load depending on the vertical amplitude using the FEA still needs improvement, lacks validation and its accuracy highly depends on the properties chosen for the virtual model [40].

Besides the above-discussed inherent limitations of laboratory studies such as the present one, the present setup has some major strengths, as it perfectly addressed the dynamic approach [41] at body temperature [42], which is not laid down in the ISO specification. The dynamic method is considered more representative of clinical conditions and is, therefore, more commonly used than the static method [42]. The matching artificial root canals individually fabricated for each instrument with their specific dimensions simulated the clinical situation better than the often-used parallel tubes. The matching artificial root canals markedly reduced the longevity of the instruments compared to testing in a parallel tube and a larger dynamic amplitude time to fracture [43]. Nonetheless, specific models with cylindrical pins or concave and convex assembly models were also suited to obtain well-controlled conditions similar to the model used in the present study [38].

## 5. Conclusions

Within the limitations of this study, it was concluded that instruments with a proprietary heat treatment resulting in blue-coloured instruments with an A_f_ temperature near the body temperature exhibited higher NCF and longer TTF values than SE and Gold heat-treated instruments when being tested in severely curved, individual matching artificial root canals at body temperature. The phase transformation behaviour and the individual-specific characteristics and design features of the instruments were decisive for cyclic fatigue resistance. Heat treatment does not automatically outweigh the file parameters, in particular, variable, regressive cross-sectional core mass reduction towards the shank.

## Figures and Tables

**Figure 1 materials-17-00827-f001:**
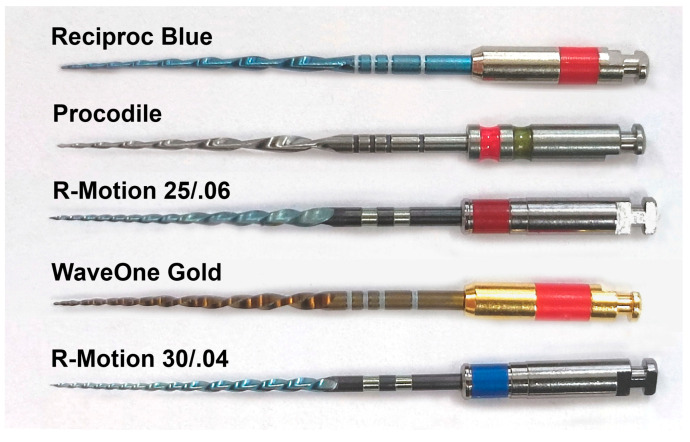
Reciprocating instruments included in the study; Reciproc Blue (RPB), Procodile (Proc), R-Motion 25/.06 (RM_06), WaveOne Gold (WOG), R-Motion 30/.04 (RM_04).

**Figure 2 materials-17-00827-f002:**
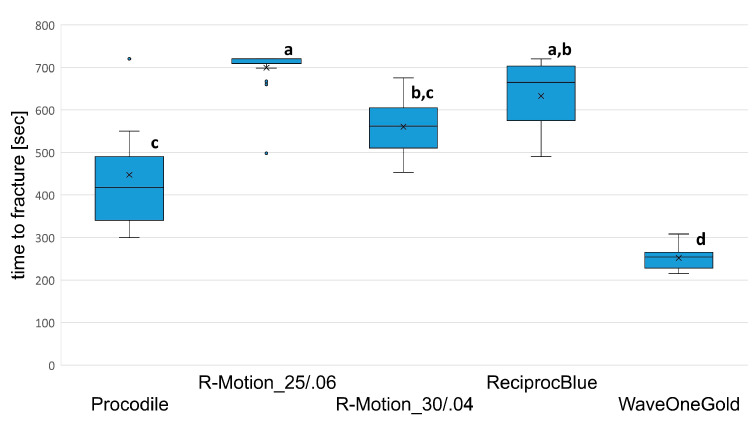
Time to fracture of the different groups with mean, median, 95% confidence interval and outliers. Bars/groups with the same alphabet letter did not differ significantly (significance level *p* < 0.05).

**Figure 3 materials-17-00827-f003:**
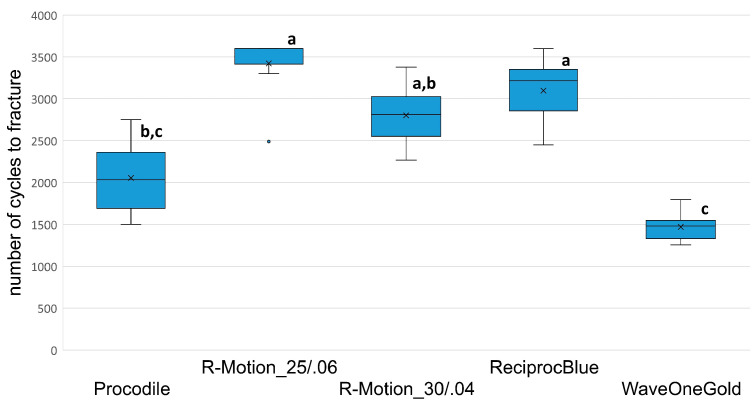
Number of cycles to failure of the different groups with mean, median, 95% confidence interval and outliers. Bars/groups with the same alphabet letter did not differ significantly (significance level *p* < 0.05).

**Figure 4 materials-17-00827-f004:**
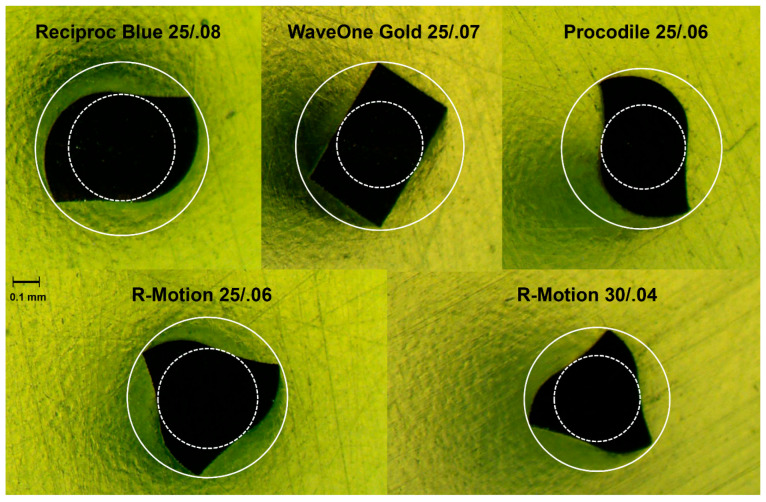
Cross-sectional characteristics of the tested instruments exactly at 6 mm from the tip. Note: different core mass (dotted line) and outer diameter (full circle) were obvious.

**Table 1 materials-17-00827-t001:** Fractures, TTF, NCF and lengths of fractured fragments.

		Time to Fracture (TTF)	Number of Cycles to Fracture (NCF)	Fracture Length
Instrument (*n* = 20 in Each Group)	Fractured Instruments (Until 720 s)	Median with Lower and Upper 95% Confidence Interval [s]	Median with Lower and Upper 95% Confidence Interval	Median with Lower and Upper 95% Confidence Interval [mm]
**Reciproc Blue ^#,†^ (RPB)**	16/20 ^b^	643 a,b [590–662]	3215 a [2951–3308]	3.55 b [2.62–3.88]
**R-Motion 25/.06 ^#^ (RM_06)**	06/20 ^a^	720 a [655–724]	3600 a [3275–3620]	2.42 a,b [2.24–3.17]
**Procodile ^#,†^ (Proc)**	18/20 ^b^	406 c [369–488]	2032 b,c [1845–2239]	2.85 a,b [2.63–2.99]
**WaveOne Gold * (WOG)**	20/20 ^c^	254 d [242–262]	1482 c [1412–1530]	2.15 a [2.17–2.48]
**R-Motion 30/.04 ^#^ (RM_04)**	20/20 ^c^	562 b,c [527–589]	2810 a,b [2634–2944]	3.03 b [2.81–3.20]

^#^ Reciproc All motion (10 cycles/s; 150°ccw, 30°cw). * WaveOne All motion (10 cycles/s; 170°ccw, 50°cw). ^†^ All tests were limited to 720 s; in groups that exceeded the max. time, 720 was used for analysis. Number of fractured instruments = chi-square test; significance level *p* < 0.05. Time to fracture, number of cycles to failure, fracture lengths = Kruskal–Wallis test with Bonferroni correction; significance level *p* < 0.05. Different superscript letters in a column indicate statistical significant differences (*p* < 0.05).

## Data Availability

The data presented in this study are available on reasonable request from the corresponding author or S.B.

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
