# Peer review of "Cyclic Fatigue of Different Reciprocating Endodontic Instruments Using Matching Artificial Root Canals at Body Temperature In Vitro"

_materials, 2024, doi:10.3390/ma17040827_

Round 1
Reviewer 1 Report
Comments and Suggestions for Authors
The manuscript investigated the cyclic fatigue of different reciprocating endodotic instruments and some valuable results were obtained and disscussed. While some issues should be clarified.
1. 5 kinds of endodontic instruments were used in this study, the materials, processing state, design, and the original pictures of the instruments should be provided. Since WaveOne Gold Primary showed rather shorter lifetime, which is due to the materials processing state or mainly due to the design? More detailed description of the instruments could help to understand the results.
2. For the artificial cannals, why use metals CoCr, not ceramics like HA or other ceramics.
3. For the fracture surface, the magnification might not be enough and it's difficult to differetiate the fracture surface details from the figure. Maybe SEM analysis could provide more information.
4. The FEA technique is mentioned and also a common technique used for lifetime prediction or analysis, could you compare the experimental results to the FEA results?
Author Response
The manuscript investigated the cyclic fatigue of different reciprocating endodotic instruments and some valuable results were obtained and disscussed. While some issues should be clarified.
- 5 kinds of endodontic instruments were used in this study, the materials, processing state, design, and the original pictures of the instruments should be provided. Since WaveOne Gold Primary showed rather shorter lifetime, which is due to the materials processing state or mainly due to the design? More detailed description of the instruments could help to understand the results.
Dear Author, thank you very much for your comments.
We added a figure showing instrument properties (longitudinal axis) in MM.
- For the artificial canals, why use metals CoCr, not ceramics like HA or other ceramics.
The artificial canals were produced in a special laser sintering approach as mentioned in MM. The ISO test were also based on test in parallel steel tubes. Thus, we decided to use CoCr that allows the special manufacturing process. Friction-free rotation was guaranteed by the use of Glycerine-oil during testing at body temperature. CoCr seems a material that is well suited for such purposes.
- For the fracture surface, the magnification might not be enough and it's difficult to differetiate the fracture surface details from the figure. Maybe SEM analysis could provide more information.
Dear reviewer, the Keyence laser scanning microscope allows a resolution of 0.1nm. Thus, it provides accurate information comparable to a SEM. Please see attached file for details.
- The FEA technique is mentioned and also a common technique used for lifetime prediction or analysis, could you compare the experimental results to the FEA results?
The idea to compare the FEA results to the results of the present study would be interesting. However, the actual FEA analyses were based on different parameters that do not match exactly to the present setup. This would represent a new study. “Cyclic fatigue of endodontic instruments – FEA versus laboratory results”

Reviewer 2 Report
Comments and Suggestions for Authors
Introduction.
This investigation is a paper that presents information for researchers in the field of endodontics. NiTi instruments are widely used due to their unique mechanical properties, such as high flexibility and pseudoelasticity. However, these instruments can fail due to cyclic fatigue, which occurs when they are subjected to repetitive cycles of loading and unloading during use. Reciprocation is a movement strategy used in endodontic instrumentation that involves alternating clockwise and counterclockwise rotation of the instrument. Compared to continuous rotation, reciprocation has been shown to reduce the occurrence of cyclic fatigue in endodontic NiTi instruments.
The aim of this study was to compare the cyclic fatigue resistance of different reciprocation instruments with different designs, tapers, sizes and heat treatments in matching artificial.
Materials and methods. This section is showed correctly according scientific methodology.
The authors must include the inclusion criteria for the selection of the types of instruments.
Results. The experimental findings showed the number of fractured instruments, the time and cycles to fracture and the fracture lengths of instruments. Also this section showed tables and figures of the results.
Table 1 can be improved.
Discussion. The present results corroborate current findings showing superior performance of heat treatments resulting in blue-coloured instruments. Heat treatments of NiTi alloys strongly biomechanic behavior favouring a different arrangement of the crystalline structure and a higher percentage of martensitic transformation
Obviously, all the instruments had a different cross-sectional design, core mass, diameter and tapers that can explain the main results of the experimental study.
Correctly, this section included an analysis of every experimental finding compared with other international studies.
This section is very long compared with the section of results of the experimental research
Conclusions. This section is correct.
References are appropiate
Conclusively, the study is not ready for publication.
Author Response
Introduction.
This investigation is a paper that presents information for researchers in the field of endodontics. NiTi instruments are widely used due to their unique mechanical properties, such as high flexibility and pseudoelasticity. However, these instruments can fail due to cyclic fatigue, which occurs when they are subjected to repetitive cycles of loading and unloading during use. Reciprocation is a movement strategy used in endodontic instrumentation that involves alternating clockwise and counterclockwise rotation of the instrument. Compared to continuous rotation, reciprocation has been shown to reduce the occurrence of cyclic fatigue in endodontic NiTi instruments.
The aim of this study was to compare the cyclic fatigue resistance of different reciprocation instruments with different designs, tapers, sizes and heat treatments in matching artificial.
Materials and methods. This section is showed correctly according scientific methodology.
Dear reviewer,
Thank you very much for your valuable comments.
The authors must include the inclusion criteria for the selection of the types of instruments.
We chose the instruments due to the fact that different cross sectional designs, heat treatments and surface treatments should be represented. Additionally, instruments that are commonly used were included. The unique parameters and designs were mentioned in “Materials and Methods” section.
We added a “justification” at the beginning of our discussion.
The study assessed cyclic fatigue resistance of various reciprocating endodontic NiTi instruments. Thus, instruments representing different designs, heat treatments and pro-cessing procedures were included, as already described. Reciproc Blue and WaveOne Gold represent the most widespread and market leading reciprocating instruments with a Blue- and Gold-wire heat treatment. Procodile is a representative of conventional SE-NiTi instruments, and R-Motion (unique “Blue-like” heat treatment) is the most recent instru-ment to be launched on the market.
Results. The experimental findings showed the number of fractured instruments, the time and cycles to fracture and the fracture lengths of instruments. Also this section showed tables and figures of the results.
Table 1 can be improved.
Table 1 looks like this due to the formatting procedure. We formatted the table differently to a better readability. Please, see the revised table in the manuscript.
Discussion. The present results corroborate current findings showing superior performance of heat treatments resulting in blue-coloured instruments. Heat treatments of NiTi alloys strongly biomechanic behavior favouring a different arrangement of the crystalline structure and a higher percentage of martensitic transformation
Obviously, all the instruments had a different cross-sectional design, core mass, diameter and tapers that can explain the main results of the experimental study.
Correctly, this section included an analysis of every experimental finding compared with other international studies.
This section is very long compared with the section of results of the experimental research
The topic is part of a controversial discussion. That requires a thorough discussion and mentioning all relevant aspects. We hope that the reviewer agrees not shortening the discussion.
Conclusions. This section is correct.
References are appropiate
Conclusively, the study is not ready for publication.
We hope that with the implemented revisions, the reviewer changes his point of view and the manuscript is suited for publication, now.
Reviewer 3 Report
Comments and Suggestions for Authors
The paper titled "Cyclic Fatigue of Different Reciprocating Endodontic Instruments Using Matching Artificial Root Canals at Body Temperature In Vitro" investigates the time to fracture (TTF) and number of cycles to failure (NCF) of various reciprocating endodontic instruments in artificial root canals. It utilizes a dynamic testing model at body temperature and statistically analyzes the results.
Positive Aspects:
Comprehensive Testing Methodology: The study employs a thorough and well-designed experimental setup, including the use of a dynamic testing model that simulates clinical conditions more realistically.
Broad Spectrum of Instruments: The paper covers a wide range of reciprocating instruments, providing a comprehensive analysis of their performance under cyclic fatigue.
Statistical Rigor: The use of robust statistical methods (Chi-Square and Kruskal–Wallis tests) enhances the reliability of the findings.
Detailed Analysis of Fracture Characteristics: The study not only examines the time to failure but also investigates the lengths of the fractured instruments and their fractographic analysis, providing a deeper understanding of the failure mechanisms.
Relevance to Clinical Practice: The study's focus on the performance of endodontic instruments in conditions that mimic clinical scenarios (like body temperature) adds significant value to its findings for dental practitioners.
Drawbacks and Suggestions for Improvement:
Limitation in Simulated Conditions: While the study simulates body temperature, it does not address other variabilities in clinical conditions, such as differences in root canal anatomy or the presence of biological fluids. Future studies could incorporate these aspects for more comprehensive results.
Comparison with Clinical Data: The study would benefit from correlating its findings with clinical data on instrument failure, which would help in validating the in vitro results in a real-world scenario.
Long-Term Durability Assessment: The study focuses on immediate cyclic fatigue resistance. Including long-term durability tests would provide a more complete picture of the instruments' performance over time.
Inclusion of Additional Instrument Parameters: Exploring other parameters like flexibility and cutting efficiency in relation to cyclic fatigue could provide a more holistic view of the instrument's performance.
Broader Range of Testing Conditions: Varying the conditions of the test, such as different angles and radii of curvature in the artificial canals, could yield more comprehensive insights into the instruments' performance under a wider range of clinical scenarios.
Overall, the study is methodologically sound and contributes valuable information to the field of endodontic treatment, particularly in the selection and use of reciprocating instruments. However, incorporating the above suggestions could further enhance its relevance and applicability.
Comments on the Quality of English LanguageThe English is quite well and it is very readable.
Author Response
The paper titled "Cyclic Fatigue of Different Reciprocating Endodontic Instruments Using Matching Artificial Root Canals at Body Temperature In Vitro" investigates the time to fracture (TTF) and number of cycles to failure (NCF) of various reciprocating endodontic instruments in artificial root canals. It utilizes a dynamic testing model at body temperature and statistically analyzes the results.
Positive Aspects:
Comprehensive Testing Methodology: The study employs a thorough and well-designed experimental setup, including the use of a dynamic testing model that simulates clinical conditions more realistically.
Broad Spectrum of Instruments: The paper covers a wide range of reciprocating instruments, providing a comprehensive analysis of their performance under cyclic fatigue.
Statistical Rigor: The use of robust statistical methods (Chi-Square and Kruskal–Wallis tests) enhances the reliability of the findings.
Detailed Analysis of Fracture Characteristics: The study not only examines the time to failure but also investigates the lengths of the fractured instruments and their fractographic analysis, providing a deeper understanding of the failure mechanisms.
Relevance to Clinical Practice: The study's focus on the performance of endodontic instruments in conditions that mimic clinical scenarios (like body temperature) adds significant value to its findings for dental practitioners.
Dear reviewer,
Thank you very much for highlighting the positive aspects. We are always glad to have such favorable comments.
Drawbacks and Suggestions for Improvement:
Limitation in Simulated Conditions: While the study simulates body temperature, it does not address other variabilities in clinical conditions, such as differences in root canal anatomy or the presence of biological fluids. Future studies could incorporate these aspects for more comprehensive results.
We agree that clinical conditions differ concerning the root canal anatomy i.e. angles of curvature, radius and position concerning the corona, middle or apical portion. However, the aim was to evaluate the cyclic fatigue with well-standardized conditions. Our model guarantees these conditions.
To evaluate different root canal morphologies can be addressed in further studies aiming at the performance of one or more instruments in dependence of the corresponding variability of the root canal anatomy.
Comparison with Clinical Data: The study would benefit from correlating its findings with clinical data on instrument failure, which would help in validating the in vitro results in a real-world scenario.
We had a section in the discussion that focused on the prevalence of instruments fractures in clinical conditions (lines 71-74).
Long-Term Durability Assessment: The study focuses on immediate cyclic fatigue resistance. Including long-term durability tests would provide a more complete picture of the instruments' performance over time.
All the instruments were claimed as single-use instruments, thus the focus was not to evaluate long-term durability tests with a possible impact of immersion in sodium hypochlorite or even sterilizing processes.
Inclusion of Additional Instrument Parameters: Exploring other parameters like flexibility and cutting efficiency in relation to cyclic fatigue could provide a more holistic view of the instrument's performance.
We completely agree that this would give a more holistic view onto the instruments performances. We discussed this aspect in the discussion section and decided not to make a multi-approach test. The recently developed and unique artificial root canal model aimed at cyclic fatigue of the tested instruments.
Broader Range of Testing Conditions: Varying the conditions of the test, such as different angles and radii of curvature in the artificial canals, could yield more comprehensive insights into the instruments' performance under a wider range of clinical scenarios.
Please, see the comment above concerning the clinical data and the relevance. The more tests the more information. Setup and design of a study cannot include all data – we decided to focus singularly on the cyclic fatigue using a unique artificial root canal model designed especially for the tested instruments guaranteeing well-standardized conditions with a defined in- and out-movement (amplitude) of the instruments – meaning a dynamic approach – at body temperature.
Overall, the study is methodologically sound and contributes valuable information to the field of endodontic treatment, particularly in the selection and use of reciprocating instruments. However, incorporating the above suggestions could further enhance its relevance and applicability.
Again, thanks for the valuable comments. We addressed all the concerns by direct comments and by revisions in the manuscript.
Reviewer 4 Report
Comments and Suggestions for Authors
The reviewer appreciates the efforts of the authors to conduct this study which has good clinical significance. The study is well-designed to achieve the objectives. The manuscript is well written however, the reviewer noticed a few errors in the manuscript that need to be revised before acceptance.
The author has used several abbreviations in the abstract section in the wrong format. Please follow the instructions of the journal to revise the abbreviation in the correct format.
The reviewer suggests the author read the abstract carefully and if possible, revise the abstract to make it simple and easily understandable for the reader
The author has used customized methodology in this study that’s why it would be better to add a methodological diagram to the manuscript
Is it essential/ informative to add the Min and Max values in Table 1?
Please format the table 1 in a better way
Please add statistical significance in the table and bar graphs. Please add in the legend the way the statistical significance is expressed in the table and figure. (e.g- groups identified with the same alphabet are statistically insignificant)
Please adjust the gridline according to the minimum and maximum values expressed.
Author Response
The reviewer appreciates the efforts of the authors to conduct this study which has good clinical significance. The study is well-designed to achieve the objectives. The manuscript is well written however, the reviewer noticed a few errors in the manuscript that need to be revised before acceptance.
The author has used several abbreviations in the abstract section in the wrong format. Please follow the instructions of the journal to revise the abbreviation in the correct format.
Dear reviewer,
Thank you for your comments. We revised the abstract accordingly to increase readability.
The reviewer suggests the author read the abstract carefully and if possible, revise the abstract to make it simple and easily understandable for the reader
The author has used customized methodology in this study that’s why it would be better to add a methodological diagram to the manuscript
We introduced our unique model recently and tried to avoid redundancies. You can find the methodology in the references – all the publications were open-access manuscripts.
Bürklein, S., Maßmann, P., Donnermeyer, D., Tegtmeyer, K., Schäfer, E. Need for Standardization: Influence of Artificial Canal Size on Cyclic Fatigue Tests of Endodontic Instruments. Appl. Sci. 2021, 11, 4950. DOI:10.3390/app11114950
Bürklein, S., Zupanc, L., Donnermeyer, D., Tegtmeyer, K., Schäfer E. Effect of Core Mass and Alloy on Cyclic Fatigue Resistance of Different Nickel-Titanium Endodontic Instruments in Matching Artificial Canals. Materials (Basel). 2021, 14, 5734. DOI: 10.3390/ma14195734
Thus, the figures and diagrams are already available. We think that a recurrent duplication is not necessary. If you insist, we can add a figure.
Is it essential/ informative to add the Min and Max values in Table 1?
Initially, we decided to give these values to duplicate the values in the table. We agree, by deleting the min/max values, the readability increases. Please, see the revised table in the manuscript.
Please format the table 1 in a better way
We deleted the min- and max- values and formatted the table in a different way to improve readability.
Please add statistical significance in the table and bar graphs. Please add in the legend the way the statistical significance is expressed in the table and figure. (e.g- groups identified with the same alphabet are statistically insignificant)
We revised the table and the graphs accordingly
Please adjust the gridline according to the minimum and maximum values expressed.
In the graphs, the min and max values and outliers were given. The format is predetermined by SPSS and adjusting the gridline in another way may not be beneficial.
Reviewer 5 Report
Comments and Suggestions for Authors
Dear authors, congratulations on your submission. The manuscript is well written, with the experimental design clearly presented, as well as the evaluation methods. Here are some considerations:
The text contains some abbreviations or changes in nomenclature that make it difficult to understand. For example, the topic of results, in which the groups are abbreviated.
Table 1, figure 1 and figure 2 present redundant results. Define only one way of presenting the results. I recommend presenting the groups in the same sequence as presented in the methodology.
Analysis of the Fractured Instruments. The results of the fractographic analysis were characterized as "striations and catastrophic failure". Therefore, the methodology presents the fractographic classification based on the literature.
"Thus, the null hypothesis was rejected" The hypothesis presented in the study is not only related to cyclic fatigue resistance.
Author Response
Dear authors, congratulations on your submission. The manuscript is well written, with the experimental design clearly presented, as well as the evaluation methods.
Dear reviewer,
Thank you very much for your comments.
Here are some considerations:
The text contains some abbreviations or changes in nomenclature that make it difficult to understand. For example, the topic of results, in which the groups are abbreviated.
The abbreviations were introduced in the abstract. We added an additional mentioning in MM. Hopefully, readability is better, now.
Table 1, figure 1 and figure 2 present redundant results. Define only one way of presenting the results. I recommend presenting the groups in the same sequence as presented in the methodology.
Table and Graph represent redundancies. Most journal recommend duplication of the values in the figures. Due to the scale and the different values obtained, the graphs do not allow visibility of the exact values, whereas the figures better show the significances. Thus, we believe that having both, table and figures were more reader-friendly. Hopefully, you agree with our statement.
Analysis of the Fractured Instruments. The results of the fractographic analysis were characterized as "striations and catastrophic failure". Therefore, the methodology presents the fractographic classification based on the literature.
Fractographic analysis of cross-sectional images of fractured instruments reveals the characteristic fatigue fracture pattern of crack initiation, propagation and catastrophic fracture with pitting. Pitting over the entire fracture surface is typical of cyclic fatigue fractures, while torsional fractures show circular wear marks on the fracture surface and pitting only near the centre of rotation [21,22].
References:
Anderson, T.L.; Anderson, T.L. Fracture Mechanics: Fundamentals and Applications, 3rd ed.; CRC Press: Boca Raton, Fl, USA, 2005;doi:10.1201/9781420058215.
Cheung, G.S.; Darvell, B.W. Fatigue testing of a NiTi rotary instrument. Part 2: Fractographic analysis. Int. Endod. J. 2007, 40,619–625, doi:10.1111/j.1365-2591.2007.01256.x.
"Thus, the null hypothesis was rejected" The hypothesis presented in the study is not only related to cyclic fatigue resistance.
We revised the hypothesis accordingly:
The null hypothesis was that cyclic fatigue resistance, i.e. TTF and NCF, is equal for all the instruments tested.
Round 2
Reviewer 1 Report
Comments and Suggestions for Authors
The manuscript has been largely improved.
Reviewer 4 Report
Comments and Suggestions for Authors
Thank you for the point-by-point reply. The revision is satisfactory. The reviewer has no additional comment.
Reviewer 5 Report
Comments and Suggestions for Authors
Dear author, I have no more questions. Thank you.